# The Sublingual Microcirculation in Critically Ill Children with Septic Shock Undergoing Hemoadsorption: A Pilot Study

**DOI:** 10.3390/biomedicines12071435

**Published:** 2024-06-27

**Authors:** Gabriella Bottari, Valerio Confalone, Jacques Creteur, Corrado Cecchetti, Fabio Silvio Taccone

**Affiliations:** 1Pediatric Intensive Care Unit, Bambino Gesuù Children’s Hospital, Scientific Institute for Research, Hospitalization, Healthcare (IRCCS), 00165 Rome, Italy; valerio.confalone@opbg.net (V.C.); corrado.cecchetti@opbg.net (C.C.); 2Department of Intensive Care, Hopital Universitaire de Bruxelles (HUB), Université Libre de Bruxelles (ULB), 1050 Brussels, Belgium; jacques.creteur@hubruxelles.be (J.C.); fabio.taccone@ulb.be (F.S.T.)

**Keywords:** critical care, hemoadsorption, microcirculation

## Abstract

**Background:** The importance of perfusion-guided resuscitation in septic shock has recently emerged. We explored whether the use of hemoadsorption led to a potential beneficial role in microvascular alterations in this clinical setting. **Methods**: A pre-planned secondary analysis of a Phase-II interventional single-arm pilot study (NCT05658588) was carried out, where 17 consecutive septic shock children admitted into PICU were treated with continuous renal replacement therapy (CRRT) and CytoSorb. Thirteen patients were eligible to be investigated with sublingual microcirculation at baseline, 24, 48, 72 and 96 h from the onset of blood purification. Patients achieving a microvascular flow index (MFI) ≥ 2.5 and/or proportion of perfused vessels (PPV) exceeding 90% by 96 h were defined as *responders*. **Results:** In 10/13 (77%), there was a significant improvement in MFIs (*p* = 0.01) and PPVs% (*p* = 0.04) between baseline and 24 h from the end of treatment. Eight patients displayed a high heterogenicity index (HI > 0.5) during blood purification and among these, five showed an improvement by the end of treatment (HI < 0.5). **Conclusions:** In this pilot study, we have found a potential association between CytoSorb hemoadsorption and a microcirculation improvement in pediatric patients with septic shock, particularly when this observation has been associated with hemodynamic improvement.

## 1. Introduction

In recent years, the critical importance of “perfusion-guided” resuscitation in the management of septic shock has been increasingly highlighted in the literature [1]. Sublingual microcirculation analysis has emerged as a valuable bedside technique for investigating the impact of septic shock on organ perfusion and assessing the efficacy of therapeutic interventions by directly visualizing microvascular blood flow.

Numerous studies, both in animal models and human subjects, have consistently demonstrated that microcirculation is significantly compromised in septic shock [2]. Moreover, the severity of these microcirculatory alterations correlates with the overall severity of the disease and serves as a predictive indicator of mortality [3]. Interestingly, vasoactive medications commonly used to treat septic shock have been associated with either a limited improvement or no discernible impact on the microcirculation [4]. Notably, cumulative exposure to vasopressors in septic shock has been associated with increased mortality [5,6].

Given these considerations, there has been growing interest in exploring the potential benefits of adjuvant therapies, such as blood purification techniques. These therapies aim to reduce the reliance on vasopressors, exert anti-inflammatory effects, and ultimately enhance organ perfusion [7]. However, our current understanding of the impact of blood purification techniques on microcirculatory resuscitation in septic shock remains limited, particularly in the context of pediatric critical care [8]. In our recent study involving children with septic shock, we observed a significant reduction in the Vasoactive Inotropic Score (VIS) and Pediatric Logistic Organ Dysfunction (PELOD-2) score at 72 and 96 h following the initiation of CytoSorb therapy, when compared to baseline. Importantly, these reductions were more pronounced in the hemoadsorption group than in a historical cohort of pediatric patients who underwent continuous renal replacement therapy (CRRT) alone [9].

Furthermore, we embarked on an exploration of whether the observed hemodynamic improvements were accompanied by corresponding changes in microcirculation during CytoSorb and CRRT therapy in the same cohort of pediatric patients. Our overarching hypothesis revolved around whether the utilization of hemoadsorption, which enables control over septic shock mediators, could potentially exert a beneficial influence, not only on global hemodynamic variables, but also on microvascular alterations within this clinical setting. 

## 2. Materials and Methods

### 2.1. Study Design

This was a pre-planned secondary analysis of a Phase-II interventional single-arm pilot study conducted in the Pediatric Intensive Care Unit (PICU) at the Bambino Gesù Children’s Hospital in Rome, Italy. The study protocol underwent review and approval by the local Ethics Committee in July 2019 (Ethics Committee Reference: N376). Informed written consent was obtained from the next of kin or legal guardian for each patient enrolled in the study [9].

The patient recruitment and data collection period spanned from July 2019 to October 2021. Eligibility screening was conducted for all patients admitted to the PICU who presented with either microbiologically confirmed or suspected septic shock. The inclusion criteria for participation in the study were as follows: 1. Body weight ≥ 10 Kgs, 2. A diagnosis of septic shock as defined by the International Pediatric Consensus Conference criteria [10], and 3. The requirement for CRRT; CRRT was indicated either due to acute kidney injury, as defined using the Kidney Disease Improving Global Outcome (KDIGO) criteria [11], or the presence of fluid overload ≥10%. The study was registered in the clinical trials database under the identifier NCT05658588.

All study procedures were conducted in accordance with the ethical standards and guidelines set forth by the responsible committee for human experimentation, including institutional and national regulations. The study adhered to the principles outlined in the Helsinki Declaration, ensuring the protection of the rights, safety, and well-being of all participating patients.

### 2.2. Data Collection and Definition

Data collection for this study was managed through an electronic case report form (eCRF). At the time of patient admission (i.e., enrolment), a comprehensive set of information was documented, encompassing demographics, comorbidities, source of admission, primary diagnoses, PELOD-2 and VIS [12,13], as well as the Pediatric Index of Mortality 3 (PIM-3) score [14]. Continuous monitoring of invasive arterial pressure was initiated at baseline, which marked the onset of therapy, and persisted for a duration of 96 h. Lactate levels, VIS, and PELOD-2 scores were assessed at multiple time points, specifically at baseline, 24, 48, 72 and 96 h following the commencement of therapy. Sublingual microcirculation assessments were conducted daily for five consecutive days starting from the initiation of blood purification. A handheld vital microscope (HVM) utilizing incident dark field microscopy imaging technology (Braedius Medical, Huizen, The Netherlands) was employed to capture five video recordings at each designated time-point. Subsequently, the three videos with the highest quality were selected for subsequent analysis, in accordance with a recent consensus [15,16]. The videos underwent a meticulous offline analysis process, utilizing dedicated software (Analysis Manager V2, Braedius Medical, Huizen, The Netherlands) [17]. Additionally, a qualitative assessment was conducted by two independent operators [16,18]. Intrao-bserver and intero-bserver variabilities are around 5% in our department [18]. 

Several critical microcirculatory parameters were computed, including total small vessel density (TVD), proportion of perfused vessels (PPV), and perfused microvascular density (PVD).

For the semi-quantitative evaluation of microcirculatory flow, the methodology described by Boerma et al. [19] was adopted. Each recorded image was divided into four equal quadrants, and flow within each quadrant was assigned a numerical score: no flow (score 0), intermittent flow (score 1), sluggish flow (score 2), and continuous flow (score 3). The microvascular flow index (MFI) was then determined based on the predominant flow type in each quadrant and subsequently averaged across all quadrants. Furthermore, the heterogeneity index (HI) of MFI was also computed [20]. Patients who exhibited an improvement in microcirculation were categorized as “***responders***”; this classification was based on the attainment of specific criteria within 96 h from the commencement of blood purification treatment, including achieving an MFI ≥ 2.5 and/or PPV exceeding 90% [2,3,19].

### 2.3. CytoSorb Therapy

A hemodialysis catheter was carefully inserted into a central vein, with the choice of internal jugular or femoral placement contingent upon the patient’s size and suitability; CRRT was carried out employing a standard hemofilter constructed from either polyarylethersulphone or ANST69. In addition, a CytoSorb (CytoSorbents Inc., Princeton, NJ, USA) adsorber was integrated into the CRRT circuit, and configured to operate in continuous veno-venous hemodiafiltration (CVVHDF) mode. The effluent dosage was maintained at 2000 mL/h × 1.73 m^2^, ensuring effective blood purification.

Within the CRRT circuit, the CytoSorb cartridge was strategically positioned in series with the hemofilter, with routine replacement scheduled for every 24 h. This arrangement was sustained for a maximum duration of 96 h. Prior to each CRRT session, the circuit underwent a thorough saline solution flush and was primed with either albumin, blood, or saline, a decision made in accordance with the clinical judgment of the attending physicians.

To manage anticoagulation, unfractionated heparin (UFH) was administered through a continuous heparin–sodium infusion, typically adjusted to maintain a post-filter activated clotting time (ACT) within the range of 160 to 180 s. In cases where contraindications to UFH were identified, regional citrate anticoagulation was judiciously employed as an alternative approach.

For patients requiring extracorporeal membrane oxygenation (ECMO) support, the placement of CRRT access and return was strategically positioned post-oxygenator and pre-pump to ensure compatibility and safety within the ECMO circuit.

#### Statistical Analysis

For descriptive data, continuous variables are expressed as mean ± standard deviation (SD) or median and interquartile range (IQR), according to their distribution; categorical variables are expressed as count (*n*) or percentage (%). To evaluate changes between two timepoints, the Mann–Whitney U test was performed. Spearman correlation was used to assess the association between microcirculation and hemodynamic parameters, both in the cohorts and in the responders’ population. Differences were considered statistically significant at *p* < 0.05. All the tests were performed using NCSS 2021 Statistical Software (2021) (NCSS, LLC., Kaysville, UT, USA).

## 3. Results

### 3.1. Study Population

We enrolled a total of 17 patients, with 13 meeting the criteria for inclusion with sublingual microcirculation monitoring. The median age of the participants was 9 years, with a median weight of 35 kg. Upon admission to the PICU, the median PIM-3 score was 98.6%. Table 1 provides a comprehensive overview of the demographic and clinical characteristics of the enrolled population at the time of PICU admission, along with the subsequent outcomes. All patients received both CRRT and CytoSorb therapy within 24 h from the onset of septic shock. The median duration of hemoadsorption therapy was 72 h, with a range of 48 to 96 h. During the treatment course, patients required a median of three cartridges.

### 3.2. Microcirculatory Changes during CytoSorb Therapy

Figure 1 displays the distribution of all microcirculatory parameters within the patient population undergoing combined therapy. No statistically significant differences were observed in the microcirculation parameters at different time points during the treatment across all patients. Among responders (10/13, 77%), a significant improvement in MFIs (*p* = 0.01) and PPVs% (*p* = 0.04) was noted between the baseline and 24 h from the end of treatment. However, no significant differences were observed for other parameters in this subgroup (Table 2).

Among the responders, 7 out of 13 patients exhibited an HI > 0.5 at various time points during therapy (20). Among these patients, 4 displayed elevated heterogeneity only during the early phase (<48 h), with a subsequent normalization of the index to <0.5 by the end of the treatment (Figure 2a). Patients with higher HI values during a later period (>48 h) were primarily non-responders in terms of MFIs and PPVs improvement (Figure 2b).

### 3.3. Relationship with Hemodynamic Variables, Vasopressor Dosage and Mortality

The correlation of sublingual microcirculatory parameters with global hemodynamic variables, lactate, a PELOD-2 score and a VIS evidenced that non-responders showed an improvement; however, this was not significant for the hemodynamic parameters, the VIS, the PELOD-2 score and lactate reduction (Table 3).

We also calculated the Spearman correlation in our cohort between the Systolic Blood Pressure (SBP), Diastolic Blood Pressure (DBP), Mean Arterial Pressure (MAP), lactate and VIS with MFIs, PPVs, TVDm and PVDm, observing in *responders* patients a weak positive correlation between MFIs and VIS (r = −0.36, *p* 0.02) and a weak negative correlation between PPVs and VIS (r = −0.39, *p* 0.01) and PVDm and VIS (r = −0.38, *p* 0.01). A weak positive correlation was also found between MFIs and SBP (r= 0.36, *p* 0.02), PPVs and SBP (r = 0.31, *p* 0.05), DBP and TVDm (r = 0.32, *p* 0.04), DBP and PVDm (r = 0.27, *p* 0.07) and MAP and PVDm (r = 0.36, *p* = 0.02). No correlations were found for any patients except for a weak correlation between PVDm and MAP (r = 0.28, *p* = 0.04), and in non-responders patients (Table 4). 

Persisting microcirculation alterations, defined as MFIs < 2.5 and HI index > 0.5, predicted mortality in our cohort in two out of three patients. In these patients who did not survive (*n* = 3), two patients showed a persisting microcirculation alteration and died from septic shock; however, the third patient who showed an improvement in their microcirculation parameters died from a cerebral hemorrhage. Figure 3 reports the time course of MFIs, PPVs, PVDm and TVDm in relationship to the time course of MAP and VIS in responders and non-responders.

## 4. Discussion

Monitoring microcirculation and tissue perfusion a priority in current critical care, especially in critical conditions such as sepsis and shock. Therefore, technologies like sublingual microcirculation analysis and other techniques that monitor organ perfusion (near-infrared spectroscopy, tissue oximetry) should ideally be used, particularly in research settings, to evaluate outcomes [21].

To the best of our knowledge, this study represents the first investigation into sublingual microcirculation patterns during hemoadsorption with CytoSorb in a pediatric population with septic shock. In our cohort, we did not observe a significant improvement in microcirculation parameters across the entire population over time. However, 77% of patients exhibited improved microcirculation by the end of the blood purification treatment. Furthermore, the significant, although weak, correlation observed between the changes in microcirculation and some hemodynamic variables suggest the presence of hemodynamic coherence in this subgroup of patients [22]. In contrast, non-responders exhibited higher mortality, despite an improvement in systemic hemodynamics [22].

A higher HI index was observed in our population, indicating significant microcirculatory heterogeneity and suggesting the severity of septic shock in this cohort. Specifically, 7 out of 13 patients exhibited high HI values (>0.5), which have been previously correlated with a high risk of mortality [20]. Among these patients, four out of seven ultimately showed a normalization of microvascular flow by the end of the treatment. Late-stage heterogeneity in the microcirculation was observed in all non-responder patients and in two of the three patients who did not survive.

The potential beneficial effects of hemoadsorption on microcirculation flow are believed to result from a direct effect related to the removal of sepsis mediators responsible for endothelial inflammation and micro-thrombosis [23]. Additionally, a secondary effect is related to the reduction in vasopressor load, ultimately improving organ perfusion in septic shock. While Zuccari et al. [8] previously investigated the impact of CytoSorb in adults with septic shock showing an improvement in TVD and PVD over time, no significant changes in MFI and PPVs during therapy were reported. In our study, we cannot definitively conclude that the improvement in microcirculation flow was solely attributable to the impact of CytoSorb due to the absence of a control group. However, we have documented the time course of microcirculation parameters and their relationship with global hemodynamics, vasopressor load, and perfusion indexes.

One previous study [24] demonstrated a lack of improvement in MFIs and FCD in children who did not survive septic shock, while De Backer et al. [3] observed that late (>48 h) microcirculation alterations were correlated with non-survival in adult patients with septic shock. Consequently, we cannot exclude the possibility that changes in microcirculation are a consequence of the natural course of sepsis and the effects of standard therapies. On the other hand, the observed microcirculation improvement in our population suggests that hemoadsorption as an extracorporeal treatment was not associated with negative effects on the endothelial barrier, such as the activation of the coagulation cascade, platelet clotting or other potential adverse effects.

This study has several limitations. First, the small sample size of our cohort may have contributed to a type-II error, i.e., the study might have been underpowered to detect significant changes in the microcirculatory variables. Nevertheless, this study was designed as an exploratory pilot study with the primary aim of detecting changes in microcirculation during hemoadsorption therapy in sepsis. These initial findings justify the need for further investigation. Second, it is worth noting that the microcirculatory alterations observed in our study population were relatively mild. This mild severity might have limited our ability to detect substantial improvements in microvascular parameters over time. Third, it is important to consider that various therapeutic interventions were administered over the course of the patients PICU stay. These interventions, including fluids, dobutamine, vasopressors, and steroids, may all have had an impact on the microcirculation and could potentially have influenced study results.

## 5. Conclusions

In this pilot study, we have found a potential association between CytoSorb hemoadsorption and a microcirculation improvement in pediatric patients with septic shock, particularly when this observation has been associated with hemodynamic improvement. These preliminary findings underscore the necessity for further research to elucidate the mechanisms underlying this effect in a larger and more diverse patient population, including control groups.

## Figures and Tables

**Figure 1 biomedicines-12-01435-f001:**
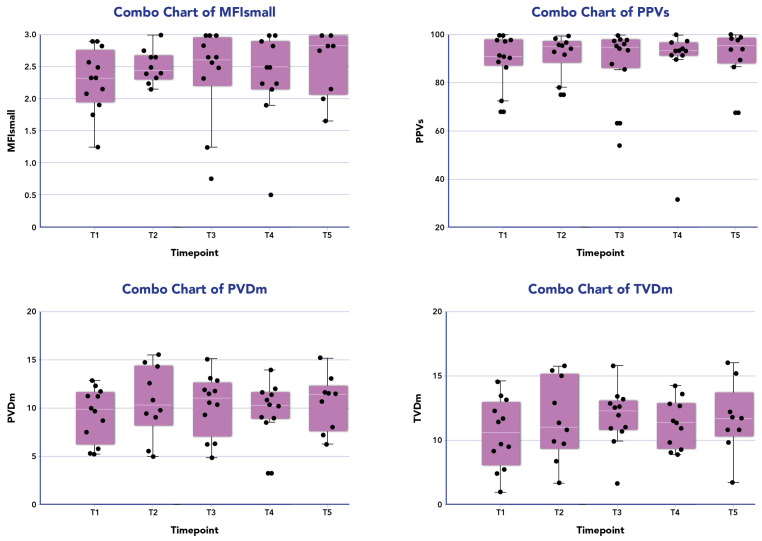
Microcirculation parameters of the patients treated with CytoSorb and Continuous Renal Replacement Therapy (CRRT) over time. The box plot (purple box) for patient investigated (black dots) are median data (interquartile range) of microcirculatory parameters for small vessels (<20 µm) of microvascular flow index (MFI) and proportion of perfused vessels (PPV), and for medium average vessels (large and small vessels) for total small vessel density (TVD) and perfused microvascular density (PVD) at the following time points: T_1_ before the onset of blood purification (BP); T_2_ 24 h from the onset of BP; T_3_ 48 h from the onset of BP; T_4_ 72 h from the onset of BP; T_5_ 96 h from the onset of BP.

**Figure 2 biomedicines-12-01435-f002:**
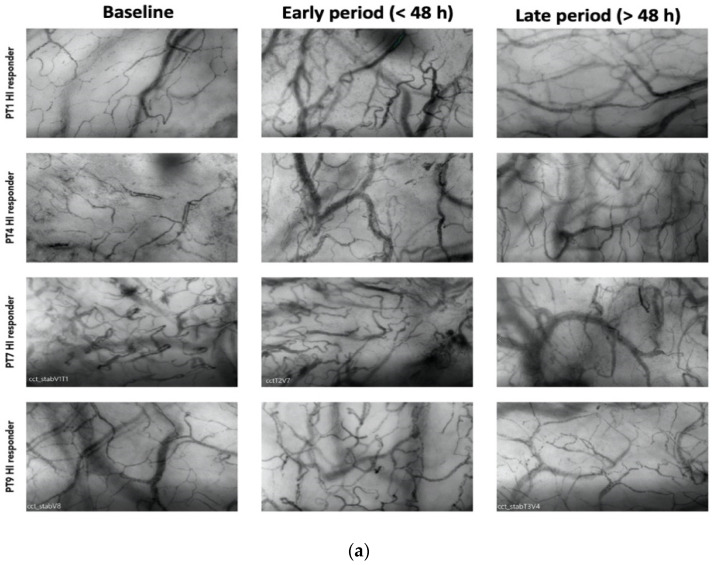
(**a**) Shows patterns of heterogenicity in microcirculation and their evolution from baseline T_1_; early period (<48 h); late period (>48 h) in those patients (4 on 7) who showed elevated heterogeneity only during the early phase (<48 h), with a subsequent normalization of the index to <0.5 by the end of the treatment (**b**) Shows patterns of heterogenicity in microcirculation and their evolution from baseline T_1_; early period (<48 h); late period (>48 h) in those patients (3 on 7) with higher HI values during a later period (>48 h).

**Figure 3 biomedicines-12-01435-f003:**
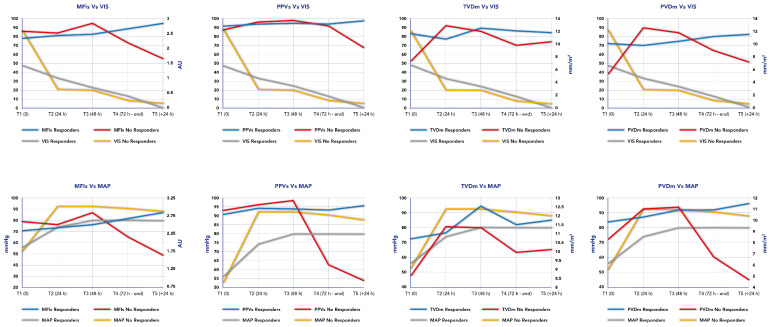
Time course of mean flow index (MFI), proportion of perfused vessels (PPV), perfused microvascular density (PVD) and total small vessel density (TVD) in relationship to time course of Mean Arterial Pressure (MAP) and Vaso-inotropic Score (VIS) in responders and non-responders.

**Table 1 biomedicines-12-01435-t001:** Demographic and clinical characteristics at PICU admission and outcome of the study population. Data are expressed as median and ranges.

Patients	13	
Gender	M: 5 (38%)	F: 8 (62%)
Age	11 [4–13]	
Source of infection:		
Primary Bacteremia	−5	
Gastronintestinal	−2	
Respiratory	−1	
Abdominal	−1	
Meningo-encephalitis	−1	
Other	−3	
**Septic shock**	**13 (100%)**	
**Vasopressors**	**13 (100%)**	
**VIS at PICU admission**	**47 [35–63]**	
**Lactate at PICU admission**	**3.9 [1.6–8]**	
**Extracorporeal support**	**13 (100%)**	
**CKRT**	**13 (100%)**	
**HP**	**13 (100%)**	
**ECMO**	**0**	
**Invasive mechanilcal ventilation**	**13 (100%)**	
**PICU LOS**	**13 [8–26]**	
**PICU Mortality**	**3 (23%)**	
**Mortality at 28 days**	**3 (23%)**	

**Table 2 biomedicines-12-01435-t002:** Median data of microcirculatory parameters for small vessels (<20 µm) of mean flow index (MFI) and proportion of perfused vessels (PPV), and for medium average vessels (large and small vessels) for total small vessel density (TVD) and perfused microvascular density (PVD) in responders and non-responders patients at different time points. Baseline: before the onset of blood purification (BP); 24 h from the onset of BP (24 h); 48 h from the onset of BP (48 h); 72 h from the onset of BP (72 h); 96 h from the onset of BP (96 h).

**MFIs AU (median)**						
	**Baseline**	**24 h**	**48 h**	**72 h**	**96 h**	***p*** **value**
**All**	2.3	2.5	2.6	2.5	2.8	0.2
**Responders**	2.3	2.5	2.4	2.8	2.8	0.01
**Non Responders**	2.7	2.5	2.7	1.3	1.1	0.2
**PPVs% (median)**						
	**Baseline**	**24 h**	**48 h**	**72 h**	**96 h**	***p*** **value**
**All**	91	24.9	94.8	93.4	94.1	0.7
**Responders**	90	95	95	93	93	0.04
**Non Resonders**	93.2	96.4	98.8	62.5	54	0.4
**TVDm mm/mm^2^ (median)**						
	**Baseline**	**24 h**	**48 h**	**72 h**	**96 h**	***p*** **value**
**All**	10.6	11.1	12.3	11.4	11.7	0.4
**Responders**	10.5	11.3	12	11	11	0.7
**Non Responders**	8.7	11.4	11.3	9.9	10.1	0.7
**PVDm mm/mm^2^ (median)**						
	**Baseline**	**24 h**	**48 h**	**72 h**	**96 h**	***p*** **value**
**All**	9.9	10.3	11	10.4	11.4	0.3
**Responders**	9.8	10.5	10.5	10.5	11	0.3
**Non Responders**	8.3	11	11.2	6.7	4.7	0.06

**Table 3 biomedicines-12-01435-t003:** Median data (interquartile ranges) of hemodynamic variables: Systolic Blood Pressure (SBP), Diastolic Blood Pressure (DPB), Mean Blood Pressure (MAP), lactate, Vasoactive Inotropic Score (VIS) and Pediatric Logistic Organ Dysfunction 2 (PELOD-2) in responders and non-responders patients at different time points. Baseline: before the onset of blood purification (BP); 24 h from the onset of BP (24 h); 48 h from the onset of BP (48 h); 72 h from the onset of BP (72 h); 96 h from the onset of BP (96 h).

**SPB mmHg (median)**						
	**Baseline**	**24 h**	**48 h**	**72 h**	**96 h**	***p*** **value**
**All**	75	100	47	100	100	0.0005
**Responders**	79	103	109	109	110	0.0045
**Non Responders**	70	125	129	115	122	0.0809
**DBP mmHg (median)**						
	**Baseline**	**24 h**	**48 h**	**72 h**	**96 h**	***p*** **value**
**All**	55	60	62	64	61	0.4225
**Responders**	55	60	62	55	60	0.6893
**Non reponders**	55	58	69	76	70	0.3711
**MAP mmHg (median)**						
	**Baseline**	**24 h**	**48 h**	**72 h**	**96 h**	***p*** **value**
**All**	55	75	81	82	84	0.0033
**Responders**	56	74	80	80	80	0.0126
**Non Responders**	53	93	93	91	88	0.3711
**Lactate mmol/L (median)**						
	**Baseline**	**24 h**	**48 h**	**72 h**	**96 h**	***p*** **value**
**All**	3.9	1.1	1.3	1.3	1.3	0.0148
**Responders**	2.8	1.3	1.3	1.3	1.3	0.0501
**Non Responders**	9	1	1.1	1.6	1.4	0.0809
**VIS (median)**						
	**Baseline**	**24 h**	**48 h**	**72 h**	**96 h**	***p*** **value**
**All**	47	32	24	13	0	0.0001
**Respoders**	47	33	24	13	0	0.0007
**Non responders**	87	21	20	8	5	0.0765
**Pelod-2 (median)**						
	**Baseline**	**24 h**	**48 h**	**72 h**	**96 h**	***p*** **value**
**All**	9	5	5	5	5	0.0031
**Responders**	8	5	5	5	5	0.0028
**Non Responders**	11	6	6	5	5	0.5536

**Table 4 biomedicines-12-01435-t004:** Spearman correlation in our cohort between Systolic Blood Pressure (SBP), Diastolic Blood Pressure (DBP), Mean Arterial Pressure (MAP), Vaso-inotropic Score (VIS), lactate (LAC) with mean flow index (MFI) and proportion of perfused vessels (PPV) and for medium average vessels (large and small vessels) for total small vessel density (TVD) and perfused microvascular density (PVD) in responders and non-responders patients.

	**Spearman Vis**		**Spearman PSYS**		**Spearman PDIA**		**Spearman MAP**		**Spearman LAC**	
	**MFIs AU** **(median)**	***p*** **value**	**MFIs Au** **(median)**	***p*** **value**	**MFIs Au** **(median)**	***p*** **value**	**MFIs Au** **(median)**	***p*** **value**	**MFIs Au** **(median)**	***p*** **value**
**All**	−0.15	0.293	0.19	0.173	−0.08	0.565	0.14	0.331	−0.31	0.362
**Responders**	0.36	0.022	0.36	0.022	−0.11	0.490	0.24	0.155	−0.08	0.617
**Non Responders**	0.38	0.181	−0.05	0.863	−0.01	0.966	0.01	0.983	−0.09	0.753
	**PPVs%** **(median)**	***p*** **value**	**PPVs%** **(median)**	***p*** **value**	**PPVs%** **(median)**	***p*** **value**	**PPVs%** **(median)**	***p*** **value**	**PPVs%** **(median)**	***p*** **value**
**All**	−0.20	0.140	0.27	0.053	−0.06	0.685	0.25	0.074	−0.27	0.054
**Responders**	−0.39	0.014	0.31	0.052	−0.12	0.478	0.28	0.089	−0.16	0.326
**Non Responders**	0.28	0.337	0.25	0.391	0.12	0.721	0.32	0.306	−0.43	0.124
	**TVD** **mm/mm^2^** **(median)**	***p*** **value**	**TVD** **mm/mm^2^** **(median)**	***p*** **value**	**TVD** **mm/mm^2^** **(median)**	***p*** **value**	**TVD** **mm/mm^2^** **(median)**	***p*** **value**	**TVD** **mm/mm^2^** **(median)**	***p*** **value**
**All**	−0.20	0.146	−0.13	0.350	0.29	0.040	0.21	0.137	−0.12	0.411
**Responders**	−0.24	0.128	−0.20	0.222	0.32	0.044	0.27	0.095	−0.06	0.698
**Non Responders**	−0.09	0.769	0.12	0.691	0.21	0.505	0.25	0.428	−0.27	0.345
	**PVDm** **mm/mm^2^** **(median)**	***p*** **value**	**PVDm** **mm/mm^2^** **(median)**	***p*** **value**	**PVDm** **mm/mm^2^** **(median)**	** *p* ** **value**	**PVDm** **mm/mm^2^** **(median)**	***p*** **value**	**PVDm** **mm/mm^2^** **(median)**	***p*** **value**
**All**	−0.24	0.077	−0.03	0.853	0.20	0.165	0.28	0.049	−0.14	0.310
**Responders**	−0.38	0.017	−0.06	0.695	0.23	0.162	0.36	0.028	−0.06	0.735
**Non Responders**	0.06	0.851	0.14	0.625	0.18	0.586	0.17	0.601	−0.34	0.233

## Data Availability

All data analyzed and discussed in the framework of this study are included in this published article. The corresponding author may provide specified analyses or fully de-identified parts of the dataset upon reasonable request.

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
