# Peer review of "The Sublingual Microcirculation in Critically Ill Children with Septic Shock Undergoing Hemoadsorption: A Pilot Study"

_biomedicines, 2024, doi:10.3390/biomedicines12071435_

Round 1

Reviewer 1 Report

Comments and Suggestions for Authors

Bottari et al have presented an interesting study showing the benefits of CytoSorb as pertaining to improvement in micrcirculation. 

This is a well designed study, however the findings have to be viewed cautiously in the absence of a control arm. Therefore I am glad that the authors have made it clear this is a pilot study.

The limitations associated with the study are well described at the end of discussion and make it clear for the reader that the absence of control group leads to certain limitations in interpretting the data and thi sstudy should only be viewed as a pilot/feasibility study. 

One minor change suggested:

In the Introduction section, I would suggest changing the wording from "exposure to vasopressors in septic shock has been linked to increased mortality" to "exposure to vasopressors in septic shock has been associated with increased mortality". Patient's with higher VIS are alse quite sicker so higher score does not necessarily mean that the vasoactive drugs are causing the mortality ( as suggested by the word "linked" ).

In the discussion section the authors can potentially discuss the devices that are available in the market that measure bucal mucosa oxygenation and circulation. We use the T-stat (by Spectros) in our pediatric ICU to great effect. The readers would want to know how the findings from a full trial can potentially impact clinical practice, and the availability of devices such as T-stat would suggest that any positive findings from a future trial would lead to actual clinical application.    

Author Response

We thank the reviewer for the comments. 

In the Introduction section, I would suggest changing the wording from "exposure to vasopressors in septic shock has been linked to increased mortality" to "exposure to vasopressors in septic shock has been associated with increased mortality". Patient's with higher VIS are alse quite sicker so higher score does not necessarily mean that the vasoactive drugs are causing the mortality ( as suggested by the word "linked"

We have changed as suggested the sentence in the introduction. 

In the discussion section the authors can potentially discuss the devices that are available in the market that measure bucal mucosa oxygenation and circulation. We use the T-stat (by Spectros) in our pediatric ICU to great effect. The readers would want to know how the findings from a full trial can potentially impact clinical practice, and the availability of devices such as T-stat would suggest that any positive findings from a future trial would lead to actual clinical application.    

We thanks the reviewer for the comments. We have included a paragraph in the Discussion section on perfusion monitoring in critical care including sublingual microcirculation analysis and tissue oximetry.

Reviewer 2 Report

Comments and Suggestions for Authors

The Sublingual microcirculation in critically ill children with septic shock undergoing hemoadsorption: a pilot study is a innovative one. However sample size is too small to consider it to be a full lenghth article. More pover in absence of comparator group it is difficult to interpret the applicability of data clinically.

I will suggest to make it a correspondence or letter to editor. Or it may be short communication

It is difficult to derive this conclusion” there is a potential beneficial effect of CytoSorb treatment on microcirculatory perfusion in septic shock patients, particularly when associated with hemodynamic improvement.” from this study

Author Response

We thank the reviewer for the comments. Considering the characteristics of the study, including the time course of microcirculation parameters and their relationship with global hemodynamic parameters, it is difficult to reduce the paper to a correspondence without losing valuable information obtained from this pilot study, which is helpful in planning future studies. Indeed microcirculation studies require granular data associated to pictures and graphs to be more explicative also for readers who are not confident with sublingual microcirculation analysis. 

We have clearly stated the limitations of our study (no control group, small sample size, etc.) multiple times throughout the main text. Following your suggestions, we have further modified the conclusions to emphasize only the "potential" association between hemoadsorption and microcirculation improvement in our cohort and the need for further studies to confirm our results.

Round 2

Reviewer 2 Report

Comments and Suggestions for Authors

Now looks better.